# Combinatorial Inhibition of Cell Surface Receptors Using Dual Aptamer-Functionalized Nanoconstructs for Cancer Treatment

**DOI:** 10.3390/pharmaceutics12070689

**Published:** 2020-07-21

**Authors:** Hyojin Lee, Tae Hee Kim, Daechan Park, Mihue Jang, Justin J. Chung, Soo Hyun Kim, Sang-Heon Kim, Kwan Hyi Lee, Youngmee Jung, Seung Ja Oh

**Affiliations:** 1Center for Biomaterials, Biomedical Research Institute, Korea Institute of Science and Technology (KIST), Seoul 02792, Korea; 213872@kist.re.kr (T.H.K.); chungjj@kist.re.kr (J.J.C.); soohkim@kist.re.kr (S.H.K.); skimbrc@kist.re.kr (S.-H.K.); kwanhyi@kist.re.kr (K.H.L.); 2Department of Biological Sciences, Ajou University, Suwon 16499, Korea; dpark@ajou.ac.kr; 3Center for Theragnosis, Biomedical Research Institute, Korea Institute of Science and Technology (KIST), Seoul 02792, Korea; mihue@kist.re.kr; 4KU-KIST Graduate School of Converging Science and Technology, Korea University, Seoul 02841, Korea; 5School of Electrical and Electronic Engineering, Yonsei University, Seoul 03722, Korea

**Keywords:** combinatorial treatment, aptamer, gold nanoconstructs, surface receptor, receptor interaction

## Abstract

Membrane receptors overexpressed in diseased states are considered novel therapeutic targets. However, the single targeting approach faces several fundamental issues, such as poor efficacy, resistance, and toxicity. Here, we report a dual-targeting strategy to enhance anti-cancer efficacy via synergistic proximity interactions between therapeutics and two receptor proteins. Importantly, we report the first finding of an interaction between c-Met and nucleolin and demonstrate the therapeutic value of targeting the interaction between them. Bispecific nanocarriers densely grafted with anti-c-Met and -nucleolin aptamer increased the local concentration of aptamers at the target sites, in addition to inducing target receptor clustering. It was also demonstrated that the simultaneous targeting of c-Met and nucleolin inhibited the cellular functions of the receptors and increased anti-cancer efficacy by altering the cell cycle. Our findings pave the way for the development of an effective combinatorial treatment based on nanoconstruct-mediated interaction between receptors.

## 1. Introduction

The treatment of many malignant diseases through the use of combinatorial drug therapies has recently attracted the attention of physicians and scientists [1,2,3,4]. Because minimizing overlapping toxicity and drug resistance is critically important for their clinical success, specific target molecules at the appropriate concentrations should be used [1,4].

Nucleolin (NCL) on the cell surface is a novel cancer target molecule for tumor therapy, due to its abundance on the surface and prominence in various cancer cells, including gastric, breast, lung, and prostate cancers [5,6,7]. In this context, anti-cancer drugs, such as immunogens, peptides, and aptamers (apts), have been intensively investigated for anti-NCL therapy [8,9,10]. Among these molecules, AS1411 is a novel nucleolin-targeted DNA aptamer and a representative anti-NCL agent that induces apoptosis through the destabilization of bcl2 mRNA by significantly inhibiting DNA synthesis and promoting the accumulation of cells in the S phase [11,12]. Although promising, AS1411 faces some fundamental limitations, including a short circulatory half-life, unexpected immune responses, and insufficient anti-cancer outcomes. These drawbacks have been mitigated using a variety of methods, such as by modifying the aptamer, conjugating the aptamer with a nanocarrier, or utilizing combinatorial treatment with other anti-cancer reagents [7].

For combinatorial treatment, the antagonism of reagents should be taken into account [13]. Many studies have reported that receptor tyrosine kinases (RTKs) are correlated with AS1411 activity. Epithelial growth factor receptor (EGFR) decreases the ability of AS1411 to stimulate macropinocytosis, whereas the inhibition of other RTKs (HER2, MET, PDGFR, IGFR-1) has little or no effect on the cellular uptake stimulated by AS1411 [12]. c-Met is an RTK member and a cell surface molecule for hepatocyte growth factor (HGF). It is considered an ideal target due to the fact that it is a key regulator of tumor development, including cancer cell invasion, growth, and vascularization [13]. For this reason, the market for anti-c-Met drugs, including anti-c-Met antibodies and anti-c-Met aptamers, has been boosted by the development of anti-nucleolin reagents [13,14,15].

When dual-targeting, nanoconstructs are actively used as a component of the delivery system [16,17]. Since the cellular capacity for the uptake of drugs and nanoconstructs is often limited [18,19], sophisticated systems are necessary to produce a synergistic effect [20]. The shape of the nanoconstructs, the drug composition, and the combination of targeting molecules should be carefully considered without compromising the fidelity of the dual treatment during strategic planning and implementation [20,21].

In this study, we propose a novel strategy through the use of a nucleolin-targeting aptamer (AS1411, N) and c-MET-targeting aptamer (c-MET apt, C) in a synergistic manner as an effective combinatorial treatment for cancer. To improve the delivery efficacy of two aptamers into gastric cancer cells (MKN-45), we synthesized star-shaped gold nanostructured (AuNS) carriers. Bi-functional AuNS or c-MET-nucleolin AuNS (AuNS-CN) alternately bind to c-Met and nucleolin on the cellular surface. More importantly, we are the first to find that the function of both nucleolin and c-MET was simultaneously depressed after the binding of AuNS-CN to a target receptor (NCL or c-MET) through interaction between NCL and c-MET. These reactions resulted in the enhancement of the anti-cancer effect in gastric cancer. The established nanocomplex increased the therapeutic efficacy of treatment by about two-fold compared with the single apt-functionalized AuNS. Using gene set enrichment analysis (GSEA) and the Molecular Signatures Database (MSigDB), AuNS-CN was found to induce exclusive enrichment patterns of gene signatures relevant to the cell cycle, representing potential unique, indirect targets for combinatorial treatment. Finally, AuNS-CN was found to induce tumor regression in vivo, while the concurrent deployment of c-Met-AuNS (AuNS-C) and nucleolin-AuNS (AuNS-N) had no significant anti-tumor efficacy. Our findings indicate that bi-functional AuNS-CN plays an important role in suppressing and eliminating the interaction between c-Met and nucleolin, ultimately resulting in cell death.

## 2. Materials and Methods 

### 2.1. Materials

Roswell Park Memorial Institute (RPMI)-1640 (Invitrogen, Carlsbad, CA, USA), fetal bovine serum (FBS) (Invitrogen), 1% antibiotic-antimycotic (AA) solution (Invitrogen), Dulbecco’s Modified Eagle’s Medium (DMEM) (Gibco, Invitrogen), 1,4-dithiothreitol (DTT) (Sigma-Aldrich, St. Louise, MO, USA), radioimmunoprecipitation assay (RIPA) buffer (89900; Pierce), bicinchoninic acid assay (23225; ThermoFisher, Waltham, MA, USA), sample buffer (125 mM Tris pH 6.8, 4% sodium dodecyl sulfate (SDS), 10% glycerol, 0.006% bromophenol blue, and 1.8% mercaptoethanol), precast protein gel (4561094; Bio-Rad, Hercules, CA, USA), primary antibodies, anti-c-Met (ab51067; Abcam, Cambridge, UK; 4560; Cell Signaling, Massachusetts, USA), anti-nucleolin (ab22578/ ab13541; Abcam), anti-HER2 (SC-08; Santa Cruz, TX, USA), anti-GAPDH (SC-47724; Santa Cruz), MTS assay solution (G3585; Promega, WI, USA), μ-Slide 8-well (ibid; München, Germany), Vectashield mounting medium with 4′,6-diamidino-2-phenylindole (DAPI) (Vector Labs), and Matrigel (BD Biosciences, NJ, USA) were used.

### 2.2. Cell Culture

All cell lines were purchased from the American Type Culture Collection (ATCC) (Manassas, VA, USA). MKN-45 and SKBR-3 cell lines were grown in Roswell Park Memorial Institute (RPMI)-1640 (Invitrogen, Carlsbad, CA, USA) supplemented with 10% FBS (Invitrogen) and 1% antibiotic-antimycotic (AA) solution (Invitrogen). A549 cells were grown in DMEM (Gibco, Invitrogen) supplemented with 10% FBS and 1% AA. All cell lines were grown at 37 °C and 5% CO_2_.

### 2.3. Aptamer-Functionalized AuNS Synthesis

Anti-nucleolin, anti-HER2, and anti-c-Met aptamers with disulfide modification at the 5′-end were purchased from IDT DNA Inc. (IDT, Coralville, IA, USA). The sequences were as follows: anti-nucleolin, 5′-(C6-S-S-C6)-TTTGGTGGTGGTGGTTGTGGTGGTGGTG-3′; anti-HER2, 5′-(C6-S-S-C6)-GCAGCGGTGTGGGGGCAGCGGTGTGGGGGCAGCGGTGTGGGG-3′; anti-c-Met, 5′-(C6-S-S-C6)-ATCAGGCTGGATGGTAGCTCGGTCGGGGTGGGTGGGTTGGCAAGTCTGAT-3′. High-performance liquid chromatography (HPLC)-purified aptamers were dissolved in Millipore water (18.2 MΩ cm) to obtain 100 µM solutions. Disulfide bonds were cleaved by adding 100 μL of 100 mM 1,4-dithiothreitol (DTT (Sigma-Aldrich, St. Louis, MN, USA) to 100 μL of aptamer solution. After 1 h, DTT was removed, and the thiolated aptamer was isolated using a Nap-5 column. The concentration of thiolated DNA was calculated by measuring the absorbance at 260 nm. The aptamer solution was added to 0.3 nM AuNS (final concentration ratio of DNA:AuNS = 1600:1) in 25 mM citrate buffer (pH 3) and incubated overnight to allow for the formation of nanoconstructs (anti-nucleolin-AuNS, anti-HER2-AuNS, and anti-c-Met-AuNS). To prepare the bifunctional nanoconstruct, two types of reduced aptamers were mixed at a 1:1 ratio (*v/v*). The mixture was then added to the AuNS solution.

### 2.4. Quantification of Aptamer-Loading Capacity of AuNS

Cy3-labeled nucleolin aptamers 5′-(C6-S-S-C6)-Cy3-TTTGGTGGTGGTGGTTGTGGTGGTGGTG-3′ and Cy5-labeled c-Met aptamers 5′-(C6-S-S-C6)-Cy5-ATCAGGCTGGATGGTAGCTCGGTCGGGGTGGGTGGGTTGGCAAGTCTGAT-3′ were used to estimate the number of aptamers on each particle. Attachment of Cy3-nucleolin and Cy5-c-Met to AuNS was performed as previously described. A total of 500 μL of fluorophore-labeled aptamer AuNS were centrifuged at 13,500 rpm for 11 min. The supernatant was removed, and the nanoconstructs were suspended in 1 mL of 50 mM HEPES buffer. This process was repeated twice to eliminate unbound aptamers. Fluorescence-labeled nanoconstruct pellets were treated with 50 μL of 25 mM potassium cyanide (KCN) overnight to dissolve the Au core of the nanoconstructs and release the Cy3 or Cy5 aptamers. The fluorescence intensity of the KCN solution was measured using a NanoDrop spectrophotometer, and the concentration of the aptamer was determined based on the intensity of the standard curve signal.

### 2.5. Western blot Analysis

The cells (5 × 10^5^ cells/mL) were cultured in a 6 well plate in complete medium for 24 h. After treating with the nanoconstructs, the cells were collected and transferred to microcentrifuge tubes. For cell lysis, radioimmunoprecipitation assay (RIPA) buffer (89900, Pierce) was added to the cells and incubated for 30 min on ice. The protein amount was estimated using a bicinchoninic acid assay (23225; ThermoFisher, Waltham, MA, USA). Briefly, an equal volume of sample buffer (125 mM Tris pH 6.8, 4% sodium dodecyl sulfate (SDS), 10% glycerol, 0.006% bromophenol blue, and 1.8% mercaptoethanol) was added to all samples, and the resulting solution was boiled for 5 min. A total of 15 μg of total protein from cells were loaded into each well of a precast protein gel (4561094; Bio-Rad, Hercules, CA, USA). After electrophoresis at 120 V for 60 min, the proteins were transferred from the gel onto a polyvinylidene fluoride (PVDF) membrane at 1 A constant current for 1 h in transfer buffer (Thermo Fisher). The blot from the transfer apparatus was removed and immediately placed into the blocking buffer (5% non-fat dry milk, 10 mM Tris pH 7.5, 100 mM sodium chloride (NaCl), and 0.1% Tween-20). After blocking for 1 h at room temperature, the membrane was incubated with primary antibodies against c-Met (ab51067; Abcam, Cambridge, UK; 4560; Cell Signaling, Beverly, MA, USA), anti-nucleolin (ab22578/ab13541; Abcam), anti-HER2 (SC-08; Santa Cruz, TX, USA), and anti-GAPDH (SC-47724; Santa Cruz) overnight at 4 °C. After incubation with the primary antibody solution, the membrane was washed twice (10 mM Tris pH 7.5, 100 mM NaCl, and 0.1% Tween-20) and incubated with horseradish peroxidase (HRP)-conjugated anti-mouse IgG (secondary antibody) diluted in 5% non-fat dry milk solution at room temperature. After 1 h of incubation, the antibody solution was removed, and the membrane was washed three times with a washing buffer (10 mM Tris pH 7.5, 100 mM NaCl, and 0.1% Tween-20). The protein bands were then detected with an enhanced chemiluminescence substrate (Atto, Tokyo, Japan). The amount of each protein in the blots was determined by counting the total number of pixels in each band (integrated density value) using ImageJ.

### 2.6. Analysis of Cell Viability by the MTS Assay 

After incubating the cells (1 × 10^4^ cells/mL) with the nanoconstructs in a 96 well plate, the cell viability was measured using MTS assay solution (G3585; Promega, WI, USA). The absorbance of the reaction solution was measured at a wavelength of 490 nm using a 96 well plate reader (Infinite 200 Pro; Tecan, Männedorf, Switzerland).

### 2.7. Immunofluorescence Staining

MKN-45 cells were seeded on an 8 well μ-Slide (ibid, München, Germany) at a density of 2 × 10^4^ cells/well for 24 h. The cells were treated with AuNS aptamers for 8 or 18 h at 37 °C, followed by fixation with 4% paraformaldehyde in PBS for 10 min. The cells were permeabilized with 0.1% Triton X-100 in PBS, blocked with 5% bovine serum albumin (BSA) in PBS at room temperature, and incubated with primary antibodies at 4 °C overnight. The cells were incubated with secondary antibodies at room temperature for 1 h and mounted with Vectashield mounting medium with 4,6-diamidino-2-phenylindole (DAPI) (Vector Labs). Images were obtained using a confocal laser scanning microscope (LSM700; Carl Zeiss, Oberkochen, Germany). Primary antibodies for c-Met (NBP2-44306; Novous, Vancouver, Canada) and nucleolin (ab22758; Abcam) were used.

### 2.8. Immunoprecipitation

Anti-nucleolin antibody was immobilized onto the surface of magnetic particles via carboxylic acid and amine coupling reaction using 1-ethyl-3-(3-dimethylaminopropyl) carbodiimide (EDC) reagent. These antibody-magnetic particles were incubated with MKN-45 cell lysate overnight at 4 °C. To determine the nonspecific protein binding of particles, bare magnetic particles (carboxylate) were incubated with the lysate. After incubation, the particles were washed twice with 1× PBS. For Western blotting, the proteins captured with anti-nucleolin magnetic particles were denatured at 95 °C using loading buffer containing DTT. After gel electrophoresis, each target protein was detected using specific antibodies. Cell lysates at the same concentration were used as a control.

### 2.9. Gene Analysis

MKN-45 cells were treated with AuNS aptamers for 24 h at 37 °C. After incubation, the cells were washed with 1× PBS and collected using trypsin-ethylenediaminetetraacetic acid (EDTA). Total RNA was extracted and used to synthesize cDNA. The samples were then applied to an Agilent 4 × 44 K Whole Human Genome microarray (Agilent Technologies, Amstelveen, Netherlands), hybridized, and washed according to the manufacturer’s instructions. The chips were scanned using an Agilent Microarray Scanner (Agilent Technologies, Amstelveen, The Netherlands).

### 2.10. MKN-45 Tumor Xenografts

All animal maintenance and in vivo experiments were conducted according to the regulations of the Institutional Animal Care and Institutional Animal Care and Use Committee of the Korea Institute of Science and Technology (project identification code: 2018-090, date: 23 November 2018). MKN-45 xenografts were generated by subcutaneously injecting 2 × 10^6^ MKN-45 cells into the dorsal flanks of female Balb/C nude mice. The experiments were initiated once the diameter of the resulting tumors was approximately 4 mm, approximately one week after injection. To evaluate the in vivo anti-cancer efficacy, 250 nM of AuNS aptamers were intratumorally injected three times at 1 week intervals (PBS for the control group). The individual tumor volume (V) was monitored every week for 4 weeks and determined using the following formula: V = (length × width^2^). At 7 days after treatment, the mice were sacrificed, and their tumors were dissected for further investigation.

### 2.11. Hematoxylin and Eosin Staining

For histological observations, the collected xenograft tumors were fixed overnight in 10% formalin (Sigma-Aldrich), embedded in paraffin, and sliced into 6 μm sections using a microtome (Leica, Wetzlar, Germany). The sections were deparaffinized in xylene and rehydrated using graded ethanol. The tumor sections were stained with H&E using standard protocols and observed under a light microscope (Olympus, Tokyo, Japan). 

### 2.12. TUNEL Assay

To evaluate apoptosis, the sections of tumors embedded in paraffin were stained with terminal deoxynucleotidyl transferase-mediated dUTP nick end labeling (TUNEL) using a commercially available TUNEL assay kit (Promega, Madison, WI, USA) according to the manufacturer’s instructions. Briefly, after routine deparaffinization and rehydration, the tumor sections were fixed with 4% formaldehyde in PBS for 15 min and washed with PBS. Proteinase K (20 µg/mL) in PBS was treated for 10 min at room temperature (RT) to digest the sections. After washing in PBS, four percent formaldehyde was applied to the sections for 5 min at RT to repeat the fixation, and the sections were immediately washed with PBS. The sections were treated with equilibration buffer for 10 min at RT before incubating with terminal deoxynucleotidyl transferase (TdT) reaction mix for 60 min at 37 °C in a humidity chamber. The reaction was terminated using 2× saline-sodium citrate stop buffer for 15 min at RT. After washing with PBS, the sections were mounted using Vectashield^®^ with DAPI (Vector Labs). For the quantitative analysis of the TUNEL assay, we adopted the apoptotic index, which is determined by the percentage of apoptotic cells counted in five randomly chosen fields at 200× magnification (n = 4 in each group).

### 2.13. Immunohistological Evaluation

To estimate angiogenesis and cell proliferation in xenograft tumors, immunofluorescence staining was performed using the rabbit polyclonal anti-von Willebrand factor (vWF at 1:100) (ab6994; Abcam, Cambridge, UK) and mouse monoclonal anti-Ki 67 (1:100) (ab238020; Abcam) as the primary antibodies. The vWF- and Ki 67-positive areas in five random fields (200× magnification) were quantified using ImageJ software (n = 4 in each group). The positive areas were quantified as the ratio of the vWF- or Ki 67-positive area to the total cell expression area.

### 2.14. Statistical Analysis

Student’s *t*-test was used to determine the statistical significance (*p*-value) for each experiment. All error bars represent the mean ± standard deviation (SD). Data were considered statistically significant with *p*-values < 0.01 (*), *p* < 0.001 (**), and *p* < 0.0001 (***).

## 3. Results and Discussion

### 3.1. Determination of the Model System

Figure 1A shows the proposed pathway for the cellular uptake of the nanoconstructs, AuNS-C and AuNS-N. As c-Met and nucleolin are often overexpressed in cancer cells, apt AuNS targeting either c-Met or nucleolin tends to recognize surface receptors. C-Met induces the internalization of AuNS-C via receptor-mediated endocytosis, while nucleolin transports AuNS-N into the cell via macropinocytosis [10,22]. To evaluate the cellular responses to the nanoconstructs, MKN-45, a gastric cancer cell line, was employed. Because of the abundance of c-Met in MKN-45, this gastric cancer cell line is a common model cell line used to screen for the therapeutic effects of anti-cancer drugs targeting c-Met [23]. Moreover, MKN-45 cells have previously shown a significant response to AS1411 compared to other gastric cancer cell lines, such as KATOIII, AGS, MKN-74, and MKN-1 [5]. Therefore, MKN-45 is an appropriate model system to verify the synergistic effect of c-Met and nucleolin combinatorial treatment. To test the specificity and therapeutic responses of the nanoconstructs, we introduced other cell lines. A549 (lung cancer) cells express c-Met and nucleolin at different levels on the cell surface compared to MKN-45 cells. Additionally, we performed a series of additional experiments with HER2, another RTK member, using the anti-HER2 aptamer to identify any forceful combination of targeting receptors. Since SKBR-3 is a representative HER2-positive cell line [22], we used this cell line as an anti-HER2 model system.

### 3.2. Synthesis and characterization of apt-functionalized AuNS

Figure 1B presents a transmission electron microscopy (TEM) image of the anisotropic AuNS, where the average size (tip-to-tip) was 50 nm. The anisotropic structure of nanoconstructs is advantageous for the effective delivery of drugs because of the large surface area compared to isotropic structures. Furthermore, the sharp tip structure reduces steric hindrance when receptors recognize their ligand [24,25]. Importantly, the toxicity of AuNS has been reported to be negligible both in in vitro and in vivo systems, which is a necessary factor for its biological application [26]. We synthesized aptamer-functionalized AuNS (nanoconstruct) with which to target the receptor on the plasma membrane. In order to attach thiolated aptamers to AuNS, a conjugation method using citric acid buffer with a low pH was employed [25]. We modified the 5′-end of three different aptamers, namely anti-c-Met, -nucleolin, and -HER2, with dithiol and grafted them onto the surface of AuNS. To synthesize the bi-functional nanoconstruct, a mixture of two aptamers with the same concentration was prepared and incubated with AuNS. During dense ligand loading on AuNS, the localized surface plasmon resonance of aptamer AuNS was found to shift to a relatively longer wavelength than that of the as-synthesized AuNS (Figure 1B,C). To calculate the amount of aptamers loaded onto AuNS, the aptamers were labeled with cyanine 3 or 5 (Cy3 and Cy5) fluoresces. The c-Met apt showed the lowest loading amount, at 222.2 (±14) per AuNS, compared to the nucleolin apt, at 402.6 (±65.9) per AuNS, and the anti-HER2 apt, at 570.1 (±25) per AuNS (Figure 1D). Although the c-Met apt only had an 8 mer difference in length (total 50 mer) compared to the 42 mer anti-HER2 apt, the total loading amount of c-Met apt exhibited a 2.5-fold lower value. This suggests that the footprint of each aptamer molecule immobilized on the surface of AuNS affects the loading density [27]. While an anti-HER2 apt has a linear structure (theoretical footprint = 1.07 nm), the c-Met apt has a secondary structure of a stem-loop (theoretical footprint = 4 nm) [14,28]. Thus, this structural feature of c-Met apt reduces its loading capacity onto the surface of the nanoconstructs. The surface charge was found to increase to ‒15 mV in PBS solution from ‒30 mV of the bare AuNS (Figure 1D). As shown in Figure 1D, all nanoconstructs had a negatively charged surface, which contributed to their stable suspension in the cell culture medium and in whole blood via repulsion between nanoparticles [26].

After the characterization of the nanoconstruct, inductively-coupled plasma-mass spectroscopy (ICP-MS) analysis was performed to measure the amount of gold inside the cells treated with three different nanoconstructs at the same concentration of aptamers. The expression levels of the receptors c-Met, nucleolin, and HER2, from MKN-45, A549, and SKBR-3, respectively, were determined using Western blotting. The validity of the experimental results was determined by comparing them against ICP-MS data. The MKN-45 cells were found to actively take up the three types of nanoconstructs, while SKBR-3 cells dramatically responded to anti-HER2-AuNS (AuNS-H) (Figure 1E). The A549 cells showed a lower cellular uptake of AuNS-C compared with the MKN-45 cells. These results indicated that the cellular uptake was noticeably dependent on the expression level of the targeting receptor, suggesting that our nanoconstruct was specific to the receptor due to the interaction between the aptamer and the receptor.

### 3.3. Therapeutic Effect of Single Aptamer-AuNS Treatment

By labeling apt AuNS with a fluorophore, the nanoconstructs were confirmed to be introduced efficiently and well distributed in the cells (Figure 2A). In the MKN-45 cells, AuNS-C and AuNS-N were located in the center of the mass around the nucleus, while AuNS-H was found near the plasma membrane with clustering. Since each nanoconstruct showed a distinctly different distribution pattern inside the MKN-45 cells, it can be hypothesized that each nanoconstruct had different therapeutic efficacies and behaviors. Therefore, a viability test was carried out after treating cancer cells with single apt-functionalized AuNS. Additionally, the therapeutic efficacy of apt AuNS was compared with that of one of the free molecules to confirm the conjugation effect of AuNS. The viability of the MKN-45 and SKBR-3 cells was found to be significantly decreased after treatment with each nanoconstruct, while the free molecules of apts did not suppress cell viability, even at the same concentrations as the aptamers (Figure 2B). Interestingly, the MKN-45 cells showed more sensitive responses to AuNS-C and AuNS-N (at a 1000 nM apt concentration) than SKBR-3 cells after 48 h of treatment.

Viability tests on the three different cell lines (MKN-45, SKBR-3, and A549) were performed to confirm the therapeutic outcome based on the distinct expression levels of each receptor at various concentrations of each nanoconstruct. After 24 h of treatment, the viability of the cells was found to decrease with an increasing concentration of c-Met and nucleolin apt AuNS, a typical dose-dependent phenomenon. Since A549 is a HER2-negative cell line, HER2-AuNS exhibited no detectable growth inhibitory effect on A549 cells. This result also supports the previous claim that these nanoconstructs have specificity to target molecules even after functionalization with aptamers on their surfaces. 

Viability and phenotypic changes were the primary evidence of the effects induced by apt AuNS. In order to further investigate the mechanism underlying the reduced viability following a single treatment of apt AuNS for an individual target, we profiled the global gene expression (Agilent, SurePrint Human Gene Expression) of MKN-45 cells in response to three single apt AuNS treatments. AuNS treatment without apt conjugation was used as a control to calculate the fold change in gene expression. Gene ontology analysis of genes up- or down-regulated by over 1.3-fold revealed that the genes related to growth suppression were significantly over-represented, suggesting that the single apt AuNS leads to cell death by disrupting the gene expression network (Figure 2D). Interestingly, although the term “negative regulation of growth” was highly ranked among the three single treatments, the sets of regulated genes were not identical. For instance, BBC3 (BCL2 Binding Component 3) was overexpressed by all three single treatments, whereas ING5 (Inhibitor of Growth Family Member 5) and GPC3 (Glypican 3) were misregulated by AuNS-H alone, indicating that combinatorial targeting could induce a synergistic effect due to the mechanisms of action dependent on the apt AuNS treatment. 

### 3.4. Combinatorial Treatment for Synergistic Effects

After a single treatment, a characteristic response of MKN-45 to c-Met or nucleolin was observed. Subsequently, a growth inhibition experiment was performed using cancer cells treated with bi-functional AuNS with two types of aptamers to evaluate their synergistic effects (Figure 3A and Appendix A). The MKN-45 cells exhibited therapeutic responses only in the combinatorial treatment with AuNS-CN at low concentrations. The growth inhibition rate of cancer cells treated with 250 nM of bi-functional AuNS (total aptamers, c-Met + AS1411) was similar to that observed with 1000 nM of the single aptamer. In addition, the therapeutic impact of the free molecule mixture of the two aptamers was almost negligible at the same concentration (250 nM) (Figure 3B). Thus, combinatorial treatment with c-Met and nucleolin aptamers using AuNS resulted in a four-fold increase in the therapeutic efficacy compared to the single aptamer treatment at a low concentration.

Following the evaluation and confirmation of the synergistic effect of AuNS-CN, we identified the location of aptamers inside the cells. The green signal for nucleolin was found to be well merged and overlapped with the red signal for c-Met (Figure 3C), indicating that the two aptamers could work at the same location in the cells. Additionally, the fluorescence signals were found to be clustered at the intracellular region. 

To determine whether the aptamers functionalized on AuNS retained their binding properties, a test to evaluate the binding of the nanoconstruct to its targeting receptor was performed using immunoblotting (Figure 3D). After incubating each apt-AuNS with MKN-45 cell lysate, the c-Met receptor was detected from both AuNS-C and AuNS-N, while nucleolin was only detected from AuNS-N. To clarify whether c-Met interacted with the anti-nucleolin aptamer or with nucleolin protein, an anti-nucleolin antibody was immobilized on the surface of a magnetic particle. For the immunoprecipitation experiment, MKN-45 cell lysates with antibody-magnetic beads were incubated overnight at 4°C. After separation and washing, the proteins captured with magnetic beads were denatured and detected using immunoblotting. Figure 3E shows that c-Met was detected with nucleolin, while the remaining proteins failed to show any interaction with the anti-nucleolin aptamer. These results demonstrated a specific interaction between c-Met and nucleolin, which is beneficial for the enhancement of the therapeutic efficiency of the combinatorial treatment with two aptamers. 

Based on these results, we hypothesized that c-Met interacts with nucleolin on the plasma membrane, which is then internalized into the cells. Thus, we anticipated that sequential treatment with AuNS-C and AuNS-N may give rise to a relatively lower therapeutic efficiency than treatment with AuNS-CN. The targeting receptors tended to disappear from the plasma membrane after pretreatment when using nanoconstructs targeting only one receptor. To verify this assumption, MKN-45 cells were incubated with either AuNS-C or AuNS-N, followed by the addition of the other nanoconstruct. Sequential treatment was found to have a less significant influence on the viability of cells compared with the bi-functional-AuNS treatment, consistent with our above assumption (Figure 3F). The treatment of the cells with a mixture of AuNS-C and AuNS-N resulted in a relatively lower cell viability compared to that observed after treatment with AuNS-CN (Figure 3F). Furthermore, ICP-MS analysis revealed that the cellular uptake of AuNS-CN was twice as high as that observed using a mixture of AuNS-C and AuNS-N (Appendix A). These results suggest that c-Met competes with nucleolin for binding sites on each apt AuNS, resulting in a less effective therapeutic outcome. 

Next, we measured the structural and stability changes of the aptamers upon interaction with the nanoconstructs (Figure 4). Circular dichroism (CD) measurement of the anti-nucleon aptamer with a G-quadruplex structure showed marked peaks at ~260 nm and ~210 nm for local maxima and ~240 nm for local minima. The anti-c-Met aptamer formed a G-loop structure, with a CD peak at ~277 nm for the local maximum and at ~245 nm for the local minimum, when dissolved in water. The results of the CD measurements suggested that the structures of the aptamers were preserved after functionalization onto the surface of AuNS (Figure 4B,C). Importantly, the bi-functional AuNS (AuNS-CN) tended to maintain its original peaks, which were well-matched with the mixtures of free aptamer molecules (anti-c-Met aptamer + anti-nucleolin aptamer), while the peaks of the AuNS-C and AuNS-N mixture became much less distinguishable (Figure 4D). These results indicate that the bi-functionalization of two aptamers to the same particle is a more effective method for ligand presentation to targeting cells compared to the mixture of single functionalized aptamers. These ligand presentation results from the CD measurements further confirm the enhanced efficacy of AuNS-CN compared to the efficacy of the AuNS-C and AuNS-N mixture in cancer cells. Taken together, these results demonstrate that AuNS-CN, the combinatorial nanostructured carriers, demonstrate enhanced therapeutic efficacy compared to a mixture of AuNS-C and AuNS-N in cancer via the targeting of c-Met and nucleolin interaction.

### 3.5. Study of the Therapeutic Mechanism of Combinatorial Treatment by AuNS-CN

Global gene expression profiles were analyzed after treatment with single apt AuNS or AuNS-CN to elucidate the transcriptional mechanism underlying the combinatorial treatment. To compare the similarity among the transcriptional profiles, we performed unsupervised hierarchical clustering. Close clustering of the two controls (i.e., mock and AuNS alone) showed that AuNS treatment alone had little effect on transcriptional change (Figure 5A). Interestingly, AuNS-CN clustered later with the agglomerated tree of the cluster among single treatments, suggesting that dual conjugation induced an additional therapeutic mechanism (Figure 5A). To compare the differentially expressed genes (DEGs) among the treatment groups, the normalized probe intensity was simply divided by the corresponding probe intensity in AuNS, using a 1.3-fold change cutoff. Using a four way Venn diagram, AuNS-CN was found to have the highest number of unique DEGs, which supports further clustering due to the unique transcriptional perturbation by the combination.

We hypothesized that the step-wise downregulation of genes by single and combinatorial treatment could explain the synergistic mechanism of the combinatorial treatment. We defined this step-wise trend as Mock:AuNS:AuNS-H:AuNS-C:AuNS-N:AuNS-CN = 5:5:4.5:4.5:4.5:4. The ratio was derived and modified from the probe intensity of BCL2 and was used as a phenotype label for GSEA. In other words, the gene expression probes were ranked in descending order of correlation with the ratio; genes with closer step-wise trends were ranked higher in the list. 

The GSEA and the Molecular Signatures Database (MSigDB) revealed that AuNS-CN induced exclusive enrichment patterns of gene signatures relevant to the cell cycle (cell cycle G1-S phase transition, nuclear chromosome segregation, DNA replication, and regulation of RNA stability) (Figure 5B). Figure 5C shows the top 50 genes in the enriched gene sets. The genes that were further downregulated by combinatorial treatment were highly likely to account for the synergistic effect of the combinatorial treatment, indicating that the genes related to the cell cycle could be unique and/or strong indirect targets of combinatorial treatment (Appendix A and Figure 5). 

### 3.6. Enhanced In Vivo Therapeutic Efficacy of Bi-Functional AuNS-CN, the Combinatorial Gold Nanoconstructs Programmed for Targeting Receptor Interaction in Human Gastric Cancer

To confirm the therapeutic efficacy of the bi-functional AuNS-CN in vivo, we used a xenograft mouse model in which MKN-45 was injected subcutaneously into the dorsal flanks of Balb/C nude mice. A week after injection, the mice whose tumor size reached 4 mm were intratumorally injected with PBS (control) or 250 nM of AuNS-CN once a week for a total of three weeks. To further analyze the effect of the AuNS-CN, groups injected with single apt-functionalized AuNSs (AuNS-C, AuNS-N) or a mixture of AuNS-C and AuNS-N (AuNS-C+AuNS-N) were added to our analysis (Figure 6). Since the intratumoral injection method is advantageous for the tumor targeting efficacy of nanoparticles [29], we applied this method to the mouse model for an accurate comparison of the therapeutic effect of bi-functional AuNS with single aptamer treatment. 

From the first day of treatment, the tumor volume measured at one week intervals only tended to decrease in the group treated with AuNS-CN over time (Figure 6B). In addition, compared to the control group, the tumor volume measured after explant was significantly reduced in the AuNS-C (*p* = 2.74 × 10^−3^) and AuNS-CN (*p* = 4.17 × 10^−4^) groups, whereas the AuNS-N and AuNS-C + AuNS-N groups did not show a significant difference (Figure 6C).

H&E and TUNEL staining (Figure 6D,E) showed that, in the groups treated with AuNS apts, cellularity was significantly reduced, while the apoptosis of cells was effectively increased compared to the control group (*p* < 0.0001). In particular, the AuNS-CN group showed an extremely higher apoptotic index value than the AuNS-C, AuNS-N, and AuNS-C+AuNS-N groups (64.36 ± 11.03% in the AuNS-CN group, 31.18 ± 3.86% in the AuNS-C group, 10.83 ± 0.82% in the AuNS-N group, and 11.47 ± 3.56% in the AuNS-C+AuNS-N group). 

Anti-c-Met and anti-nucleolin apt are known for their ability to inhibit cancer cell proliferation and their anti-angiogenic effect [13,30]. For these reasons, we performed immunofluorescent staining for vWF (green) and Ki 67 (red) to investigate the effect of the AuNS-CN on tumor angiogenesis and proliferation (Figure 6F). The AuNS-CN group showed the lowest vWF+ cells ratio of 0.56% (±0.65), followed by the AuNS-C group 2.49% (±2.22), AuNS-C+AuNS-N 4.11% (±5.11), AuNS-N 6.69% (±2.62), and the control 21.02% (±17.19) (Figure 6G). The Ki 67+ cell ratio also showed the lowest value for the AuNS-CN group 0.65% (±0.19). Compared to the AuNS-CN group, single apt-functionalized AuNSs showed approximately three times higher and a mixture of AuNS-C and AuNS-N showed approximately seven times higher expression (Figure 6H).

The results for apoptosis, anti-angiogenesis, and anti-proliferation indicated that AuNS-CN had a high therapeutic efficacy, which tended to be similar to that observed in vitro. In particular, the results of the AuNS-CN group showed higher therapeutic efficacy than the AuNS-C+AuNS-N group, reaffirming the hypothesis that c-Met interacts with nucleolin on the plasma membrane, which is then internalized into the cells, suggesting that combinatorial targeting induces a synergistic effect. Interestingly, in the AuNS-N and AuNS-C+AuNS-N groups, the apoptosis, anti-angiogenesis, and anti-proliferation effects were higher than those in the control group. However, the tumor volume did not show a significant difference from the control. These results were expected to be due to the fact that the AuNS-N and AuNS-C+AuNS-N groups have an apoptotic effect, but have a narrow range of efficacy. Indeed, in xenograft tumors, the therapeutic efficacies, including apoptosis, anti-angiogenesis, and anti-proliferation, were found to decrease in areas far from the treated areas. Nevertheless, the AuNS-CN group showed a high tumor volume reduction rate, in addition to apoptosis, anti-angiogenesis, and anti-proliferation effects. Taken together, we concluded that the bi-functional AuNS-CN, the combinatorial gold nanoconstructs programmed for targeting c-Met and nucleolin interaction, was able to effectively control cancer progression compared to AuNS-C and AuNS-N co-treatment.

## 4. Conclusions

The overexpression of cell surface receptors is a key factor in cancer progression and metastasis. Moreover, receptor crosstalk has significant implications for drug resistance. c-Met is one of the most valid therapeutic targets against different types of cancer, while nucleolin is also a well-known and promising target for cancer treatment due to its abundance in cancer cells. Despite the significance of targets in cancer therapy, there has been no previous evidence demonstrating an interaction between c-Met and nucleolin via a nanoconstruct. In this study, we demonstrated that c-Met and nucleolin interact with each other at the molecular level and are thus ideal combinatorial targeting partners for use in cancer treatment. Nucleolin is a member of the HGF receptor family; therefore, it responds to HGF stimulation in the absence of c-Met [31]. Based on our finding that nucleolin binds to c-Met, it is possible that nucleolin acts as a co-receptor for the c-Met signaling pathway. 

It has been previously reported that limited cellular uptake of nanoconstructs affects the efficacy of drug delivery [18]. These studies have also shown that the expression levels of the receptors can be a limiting factor for the cellular capacity of the receptor-mediated uptake of nanoconstructs. Therefore, a balanced and optimum combination of targeting receptors is needed prior to drug delivery. In our study, sequential treatment with nanoconstructs resulted in relatively lower therapeutic efficacy compared to simultaneous drug treatments. Since the uptake machinery of cells is limited, it is possible that the interaction between the nanoconstruct and the cell gradually diminished with changes in cellular metabolism. This highlights the need for appropriate combinatorial receptor co-targeting for an effective drug delivery with an enhanced therapeutic efficacy. 

Importantly, the aptamers on the surface of AuNS are more stable under physiological conditions than free aptamer molecules. The negatively charged aptamer AuNS recruits salt ions, resulting in a high local salt concentration around the particle surface [32]. As a result, the nuclease enzymatic activity decreases at the AuNS surface, which contributes to an improved aptamer stability in the body. The immobilization of aptamers onto the surface of AuNS at a high density was found to improve the anti-cancer effect, as well as increase the targeting efficacy, simultaneously, for the application of therapeutic aptamers in vivo. 

In summary, this study aimed to establish a bi-functional AuNS-CN nanoconstruct-based cancer therapy using the simultaneous c-Met and nucleolin targeting approach. We demonstrated that AuNS-CN synergistically enhanced the anti-cancer therapeutic efficacy without any noticeable compromises compared to treatment with the AuNS single molecule. The enhanced therapeutic efficacy appeared to originate from the proximity between the c-Met and nucleolin targeted by AuNS-CN. This enhanced performance of the AuNS-CN therapy was confirmed through additional experiments, wherein the enrichment of the nucleolin component of the AuNS-CN on the membrane following AuNS-CN binding to c-Met resulted in a better inhibition of nucleolin activity, and vice versa. Transcriptome analysis showed that AuNS-CN exhibited exclusive enrichment patterns of gene signatures related to the cell cycle. To the best of our knowledge, this synergistic AuNS-CN cancer therapy is the first demonstration of the inhibition of the interaction between c-Met and nucleolin. However, despite the novel finding that c-Met and nucleolin are synergetic partners and have potential applications as a cancer treatment, further study is needed to elucidate the interaction between the two receptors. We anticipate optimized combinatorial treatments targeting receptor interaction to promote the development of new solutions with which to overcome drug resistance and enhance the therapeutic efficacy of treatments against aggressive diseases.

## Figures and Tables

**Figure 1 pharmaceutics-12-00689-f001:**
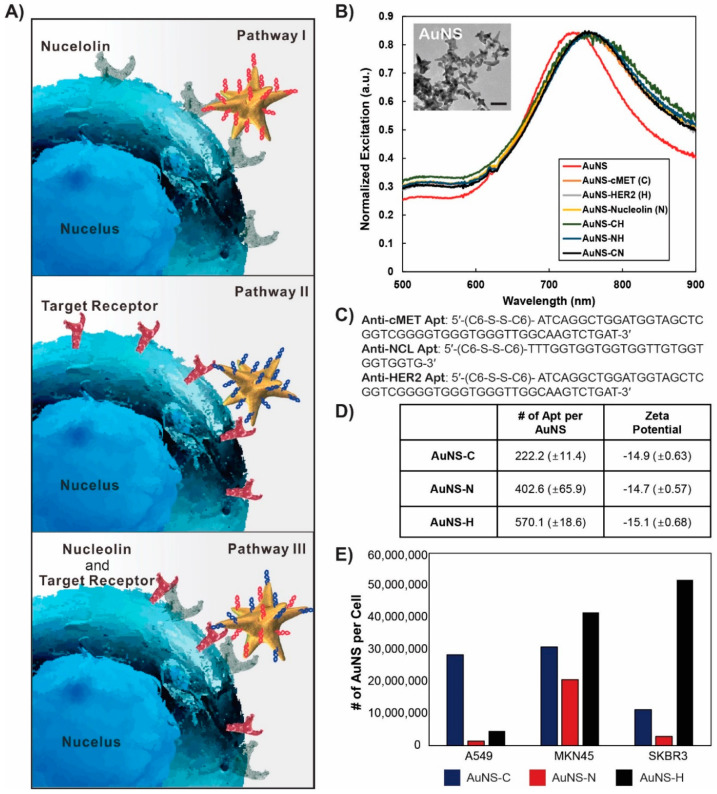
Schematic diagram of the combinatorial treatment and the characterization of the nanoconstructs. (**A**) The main mechanism of interaction between the targeting receptor and nanoconstructs. (**B**) Changes in surface plasmon resonance after functionalization of gold nanostructures (AuNS) with aptamers (apts). Scale bar = 50 nm. (**C**) The sequence of aptamers used in this study. (**D**) The amount of aptamers per AuNS and surface charge. (**E**) ICP-MS measurement for the amount of AuNS in the cells. The specific cellular uptake depending on the targeting molecules. All of the cells expressed c-MET and nucleolin on the plasma membrane (expression level of c-MET: MKN-45 > SKBR3 > A549/Nucleolin: MKN-45 > A549 = SKBR3). SKBR-3 and MKN-45 are HER2-positive cell lines, while A549 is a HER2-negative cell line.

**Figure 2 pharmaceutics-12-00689-f002:**
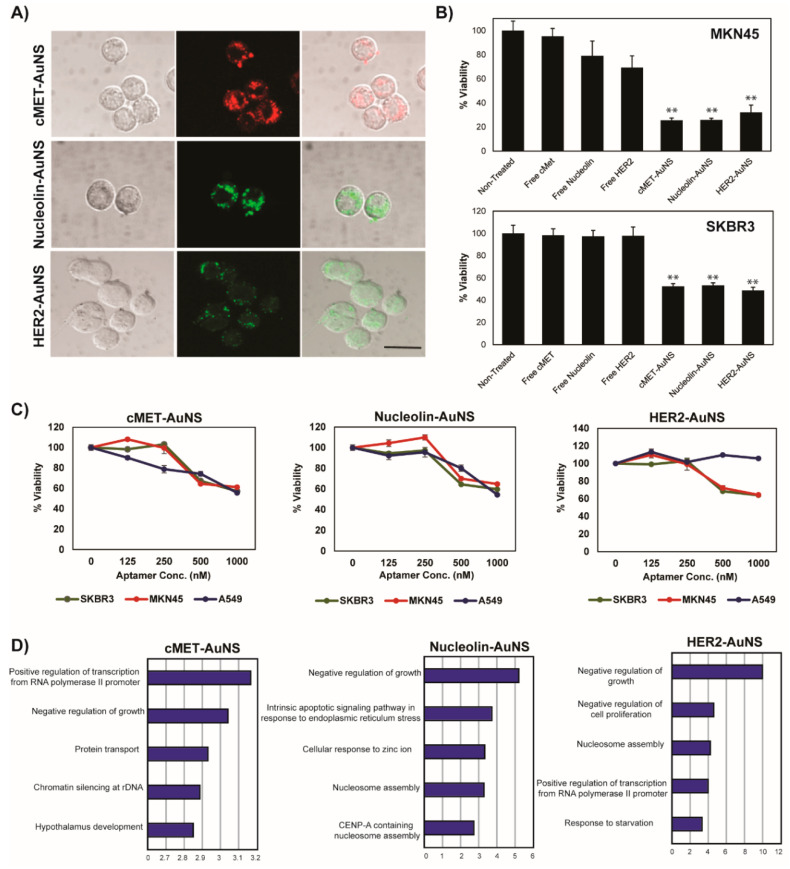
A single treatment of cells with c-MET-AuNS, nucleolin-AuNS, and HER2-AuNS. (**A**) Fluorophore-labeled nanoconstructs in MKN-45 observed with confocal microscopy. The fluorescence signals are located inside the cells, confirming the effective cellular uptake of each nanoconstruct; scale bar = 20 μm. (**B**) Comparison of the therapeutic effect between free aptamer molecules and nanoconstructs. (**C**) Result of the viability tests of SKBR-3, MKN-45, and A549 after treating the cells with aptamer-functionalized AuNS. Conc. means the concentration of aptamers which were used in cell treatment. The concentration of AuNS: anti-c-Met = 4.5 nM, anti-nucleolin = 2.5 nM, anti-HER2 = 1.8 nM. (**D**) After 24 h of treatment with 500 nM nanoconstruct in MKN-45, the genetic expression was analyzed. * *p* < 0.01, ** *p* < 0.001, or *** *p* < 0.0001 based on *t*-tests.

**Figure 3 pharmaceutics-12-00689-f003:**
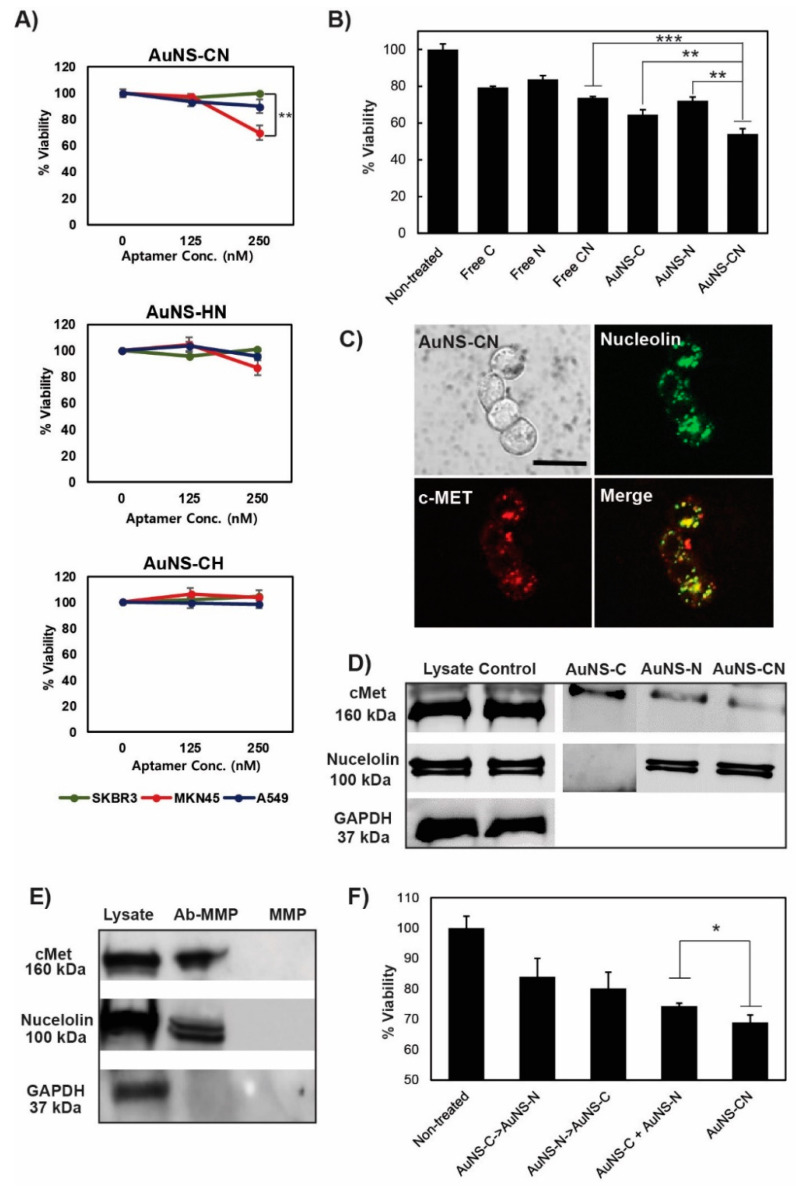
Combinatorial treatment of cells with the bi-functional-AuNS. (**A**) Cell viability using MTS. The three cell lines were treated with nanoconstructs that were functionalized by a different combination of aptamers. CN: AuNS functionalization with anti-c-MET and -nucleolin aptamer, HN: AuNS functionalization with anti-HER2 and -nucleolin aptamer, and CH: AuNS functionalization with anti-c-MET and -HER2 aptamer. (**B**) The result for viability after cells treatment with 250 nM free aptamer or 250 nM aptamer-functionalized AuNS (AuNS Conc. 1.125 nM for c-MET, 0.625 nM for nucleolin, 1 nM for CN, respectively). The enhanced therapeutic effect of aptamers by co-immobilization on the AuNS (AuNS-CN). The free aptamer mixture was not able to decrease the cell viability. (**C**) Clustered distribution of c-Met and nucleolin inside the cells after treatment with AuNS-CN; scale bar = 20 μm. (**D**) Immunoblotting results showing that c-Met binds to both AuNS-C and AuNS-N. Bi-functional AuNS (AuNS-CN) showed almost the same level of binding efficacy as AuNS-N. (**E**) Immune precipitation upon using the anti-nucleolin antibody to determine the interaction of nucleolin with c-Met. Ab-MMP is an antibody-functionalized micromagnetic particle (MMP), and MMP is a bare particle (without antibody). The immunoblotting data shows that nucleolin interacts with c-Met. (**F**) Cell viability after the sequential treatment or co-treatment of cells with apt AuNS.

**Figure 4 pharmaceutics-12-00689-f004:**
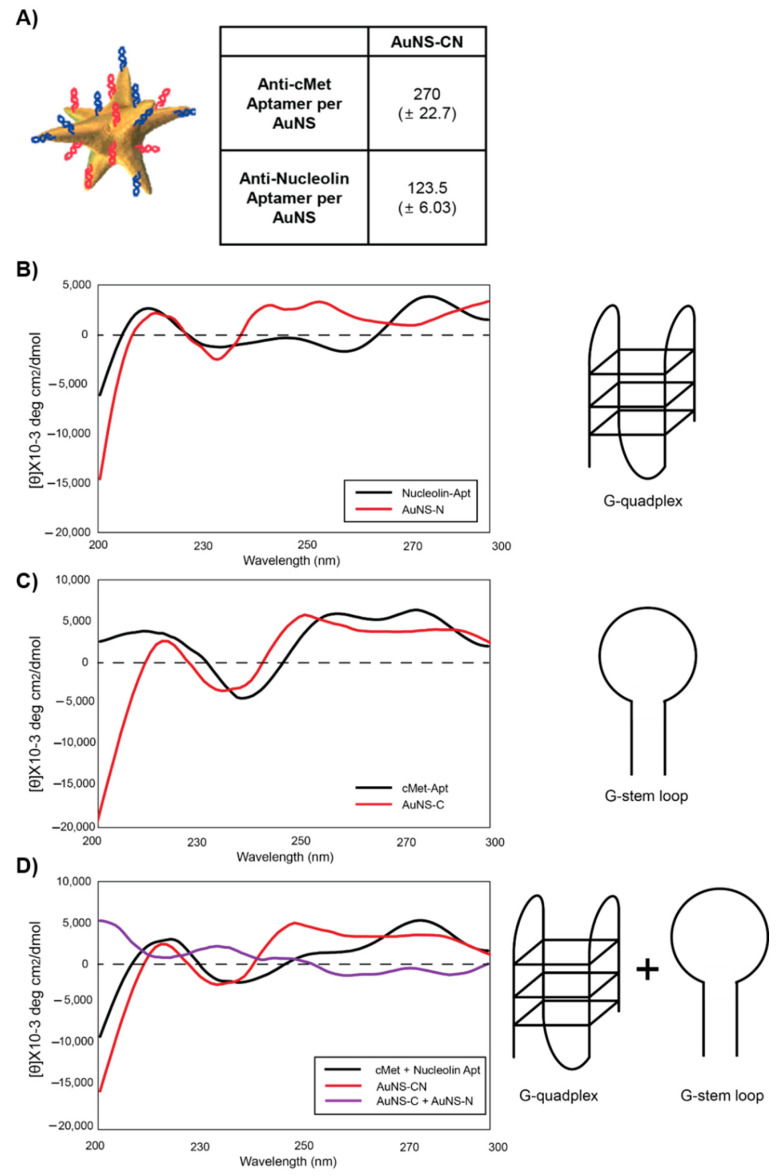
Circular dichroism (CD) measurement of the aptamer-functionalized AuNS. (**A**) The amount of aptamers per AuNS-CN and surface charge. (**B**) CD measurement of AuNS-N. (**C**) CD measurement of AuNS-C. (**D**) CD measurement of AuNS-CN.

**Figure 5 pharmaceutics-12-00689-f005:**
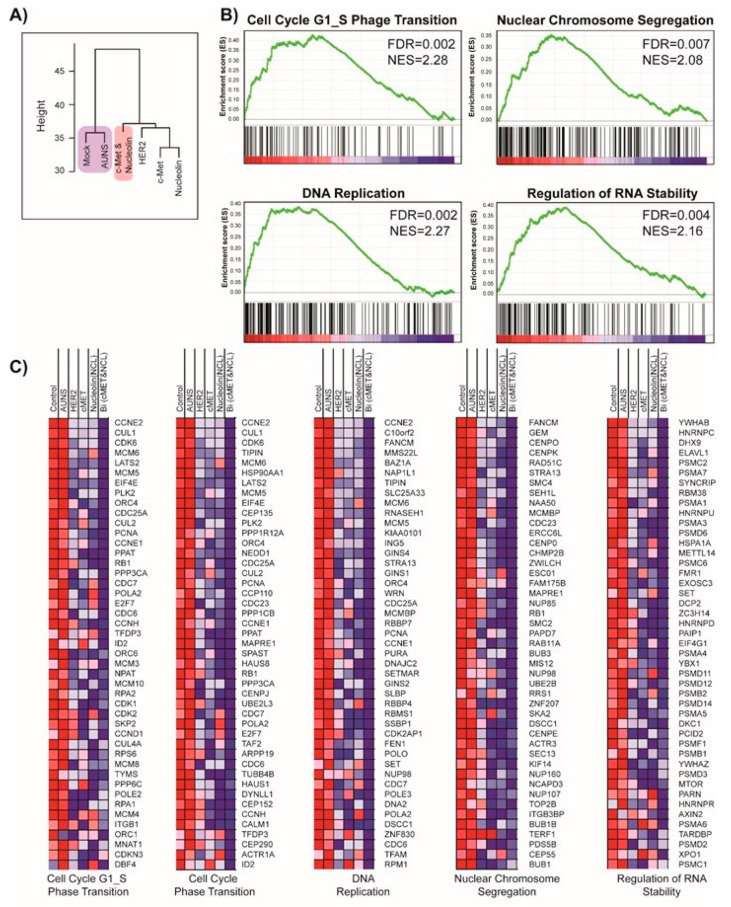
Therapeutic mechanism of combinatorial treatment by AuNS-CN. (**A**) Genetic clustering results after cell treatment with each nanoconstruct. (**B**) Gene set enrichment analysis (GSEA) analysis showing the enrichment pattern of the gene signature related to the cell cycle. FDR = False Discovery Rate, NES = Normalized Enrichment Score (**C**) Top 50 biomarkers related to the enrichment pattern of the gene signature of the cell cycle induced by AuNS-CN.

**Figure 6 pharmaceutics-12-00689-f006:**
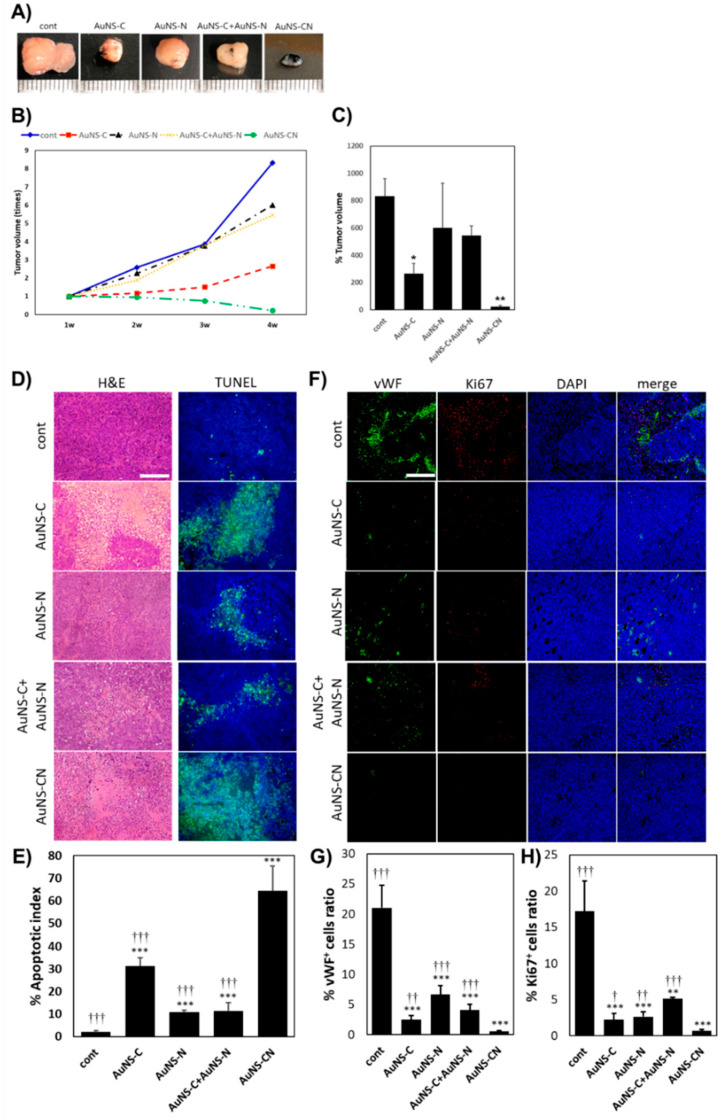
AuNS apts inhibited the growth of MKN-45 xenograft tumors in vivo and enhanced the in vivo therapeutic efficacy of combinatorial gold nanoconstructs targeting receptor interaction, AuNS-CN. (**A**) Excised xenograft tumors treated with PBS (control), AuNS-C, AuNS-N, AuNS-C+AuNS-N, and AuNS-CN three times at one week intervals in Balb/c nude mice (n = 4) injected with 2× 106 MKN-45 cells. (**B**) The ratio of tumor volume at the indicated time points to initial tumor volume, in each group. (**C**) Rate of excised tumor volume (4 w) to initial tumor volume in each group. (**D**) Representative images of H&E and TUNEL staining at excised xenograft tumors treated with PBS (control) and AuNS apts (AuNS-C, AuNS-N, AuNS-C+AuNS-N, AuNS-CN). In the TUNEL staining images, the green area indicates the distribution of apoptotic cells. Scale bar = 300 µm. (**E**) Quantification of the apoptotic index (%) of each group. The apoptotic index is the ratio of TUNEL-positive nuclei to total nuclei. (**F**) Representative images of immunofluorescence-stained anti-von Willebrand factor (vWF) and Ki 67. Scale bar = 200 µm. Quantification of: (**G**) vWF-positive expression areas (%). (**H**) Ki 67-positive expression areas (%) in the area of total cells to confirm angiogenesis and tumor cell proliferation in xenograft tumors, respectively. * *p* < 0.01, ** *p* < 0.001, or *** *p* < 0.0001 compared with the control group. † *p* < 0.01, †† *p* < 0.001, or ††† *p* < 0.0001 compared with the AuNS-CN group.

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
