# Peer review of "Combinatorial Inhibition of Cell Surface Receptors Using Dual Aptamer-Functionalized Nanoconstructs for Cancer Treatment"

_pharmaceutics, 2020, doi:10.3390/pharmaceutics12070689_

Round 1
Reviewer 1 Report
The manuscript “Combinatorial Inhibition of Cell Surface Receptors using Dual
Aptamer-functionalized Nanoconstructs for Cancer Treatment” by Lee et al. reports that aptamers with specificity for cMet receptor and Nucleolin combined on the same gold nanoparticles can inhibit the growth of the MNK-45 gastric carcinoma in immunocompromised mice better than the two single aptamers on gold nanopartlcles. Before the in vivo tests the nanoassemblies were characterized for their physical (circular dicroism) and their specificity (immunoblot, immunoprecipitation) and anti tumor cell in vitro activity.
Many data are presented, but they are not well organized and in some cases they are not convincing or not well interpreted and some controls are missing.
For this reasons, reading this manuscript was very hard: too many acronyms and sometimes not reported clearly. For example Fig S3, third panel. I think the title is not complete. Legends of Fig. S3: the terms of the different reagents should be the same as the other parts of the manuscript.
Fig. 1C: anti-cMET aptamer and then Fig. 2C: cMET. I guess that the authors refer to the same materials.
Synthesis of Apt-Functionalized AuNS: to what measure are referred the numbers of 222 and 402 of the aptamers on the gold nanoparticles? Only looking at the table (Fig. 1C) the reader understand that it is the number of apt conjugated on the gold nanoparticles. This information is missing in the text.
Fig. 2B: the control with gold nanoparticles alone (AuNS) is missing. The amount of the different nanoparticles is not reported in the legend of the Fig (I do not know if it is in the text, but I already went on the text so many times, that I cannot go further because of the lack of time!) For SKBR3: it is rather surprising that the HER2 aptamer has no effect.
Regarding the in vivo results it would be more significant to show the inhibition of the tumor growth in the main manuscript and not in text. The reverse for the data about vWF and Ki67, if the authors do not want to maintain in the main text, because otherwise there are too many data there.
Legends should be more detailed.
In the Supplementary (and possibly also in another point) I do not agree with the term “dramatically” in Fig S1. For ISO standard a cell viability around 70% in respect to controls is still accepted as cytocompatibility of the pharmaceutic component, and not as a clear cytotoxic effect (ISO 10993-5:2009—Biological Evaluation of Medical Devices—Part 5: Tests for In Vitro Cytotoxicity. Available online: https://www.iso.org/standard/36406.html.
Moreover, the English is not that good and it should improved.
Also the references are not so up to date.
the same ref repeated twice:
28. Eder, J. P., G. F. Vande Woude, S. A. Boerner, and P. M. LoRusso. "Novel Therapeutic Inhibitors of the CMet Signaling Pathway in Cancer." Clin Cancer Res 15, no. 7 (2009): 2207-14.
- Eder, Joseph Paul, George F. Vande Woude, Scott A. Boerner, and Patricia M. LoRusso. "Novel Therapeutic Inhibitors of the C-Met Signaling Pathway in Cancer." Clinical Cancer Research 15, no. 7 (2009): 2207-14.
Minor errors:
page 5, line 4 MKN-46 à MKN-45
MKN45 or MKN-45?
Legend Fig. S5: it must corrected to Balb/c mice
These are just examples of mistakes, since I went through to what I remember after reading the paper, but is suggestive of the little accuracy and thus little respect for the reviewers.
So the manuscript has to be resubmitted with great changes and better accuracy, so that it will be easy to get the message. Otherwise it cannot be accepted for publication
Author Response
Point-By-Point Response Letter (pharmaceutics-853310)
We are pleased to return the draft of the revised manuscript “Combinatorial Inhibition of Cell Surface Receptors using Dual Aptamer-functionalized Nanoconstructs for Cancer Treatment” by Lee et al., for consideration of publication in Pharmaceutics, based on the responses to the Reviewer 1 and 2. They found our work of interest and significance in this field, and we received an essentially positive response regarding our manuscript. We are pleased to add a number of substantial revisions requested by the Reviewers that address the concerns and comments in detail, and we believe that we have now supported our model with extensive and detailed data.
The main point of the paper is the dual-targeting strategy to enhance anti-cancer efficacy via synergistic proximity interaction of therapeutics with two receptor proteins. We have provided most of the extensive prepared additional revision requested to more vigorously address the list of mechanistic questions suggested by Reviewers (please see revised Fig. 6, Fig S1 and Fig S3). We believe that this supports the therapeutic value of targeting the cross-talk between c-Met and nucleolin by the bispecific nanocarrier densely grafted with anti-c-Met and –nucleolin aptamer increasing the local concentration of aptamers at the target sites. In the following, we addressed all the comments made by the reviewers on our manuscript (pharmaceutics-853310) point by point. In response to each raised point, we explained the corresponding changes we have made in the revised manuscript (highlighted in yellow).
Reviewer #1
Comments and Suggestions for Authors:
The manuscript “Combinatorial Inhibition of Cell Surface Receptors using Dual Aptamer-functionalized Nanoconstructs for Cancer Treatment” by Lee et al. reports that aptamers with specificity for cMet receptor and Nucleolin combined on the same gold nanoparticles can inhibit the growth of the MNK-45 gastric carcinoma in immunocompromised mice better than the two single aptamers on gold nanopartlcles. Before the in vivo tests the nanoassemblies were characterized for their physical (circular dicroism) and their specificity (immunoblot, immunoprecipitation) and anti tumor cell in vitro activity.
Many data are presented, but they are not well organized and in some cases they are not convincing or not well interpreted and some controls are missing.
For this reasons, reading this manuscript was very hard: too many acronyms and sometimes not reported clearly.
[Comment #1]
For example Fig S3, third panel. I think the title is not complete. Legends of Fig. S3: the terms of the different reagents should be the same as the other parts of the manuscript.
[Response #1]
We apologize for the incomplete information and the confusion in FigS3. We have changed the term and included the new complete information in Fig. S3 in the revised Supplementary Materials on page 3 as follows:
Revised Supplementary Materials
Figure. S3. Gene ontology analysis for overlapped or unique differentially expressed genes between AuNS-N and AuNS-CN treatment. The top GO term for shared DEGs between AuNS-N and AuNS-CN treatment was apoptosis-related, whereas the unique term with AuNS-CN treatment was phosphorylation-associated signaling pathway. X-axis represents −log10(p-value)
[Comment #2]
Fig. 1C: anti-cMET aptamer and then Fig. 2C: cMET. I guess that the authors refer to the same materials.
[Response #2]
We really apologize for the confusion. We have changed the material labeling to “cMet-AuNS, Nucleolin-AuNS and HER2-AuNS in Figure 2C and D as described in the manuscript to avoid the confusion (Figure 2 on page 9). However, we have remained anti-cMet aptamer, anti-Nucleolin aptamer, and anti-HER2 aptamer to emphasize the characterization of its conjugation to AuNS in Figure 1.
Revised manuscript
[Comment #3]
Synthesis of Apt-Functionalized AuNS: to what measure are referred the numbers of 222 and 402 of the aptamers on the gold nanoparticles? Only looking at the table (Fig. 1C) the reader understand that it is the number of apt conjugated on the gold nanoparticles. This information is missing in the text.
[Response #3]
We really appreciate Reviewer #1 for the constructive comments. We have included the additional information regarding the number of apt conjugated on the gold nanoparticles in the revised manuscript on page 6, line 41~43 as follows:
Revised manuscript
“The c-Met apt showed the lowest loading amount, at 222.2 (±14) per AuNS, compared to the nucleolin apt, at 402.6 (±65.9) per AuNS, and the anti-HER2 apt, at 570.1 (±25) per AuNS (Fig. 1C).”
[Comment #4]
Fig. 2B: the control with gold nanoparticles alone (AuNS) is missing. The amount of the different nanoparticles is not reported in the legend of the Fig (I do not know if it is in the text, but I already went on the text so many times, that I cannot go further because of the lack of time!) For SKBR3: it is rather surprising that the HER2 aptamer has no effect.
[Response #4]
We thank Reviewer #1 for the critical comments. The control with gold nanoparticles alone (AuNS) was missing in the experiment due to the particle stability issue. The bare AuNS is capped HEPES molecules that have a negative charge, that allows stable suspension in D.W through by repulsion forces between nanoparticles. However, in cell culture medium containing various ion molecules, the AuNS should be aggregated due to decrease repulsion force while aptamer (strong negative charge)-functionalized AuNS is stably suspension in media. Therefore, the effect of bare AuNS on cells cannot be accurately observed.
Importantly, the previous study clearly showed that the toxicity of AuNS with PEG was negligible both in vitro and in vivo. (Nanomedicine. Teri W. Odom et al. 2015, 11(3): 671–679) Thus, we did not add the AuNS experiment to our manuscript. However, the biocompatibility of materials is important for biological application, thus we mentioned the toxicity of AuNS and added this reference in the revised manuscript. (line 22~24, page 6, reference 26)
Revised manuscript
“Importantly, the toxicity of AuNS has been reported to be negligible both in in vitro and in vivo systems, which is a necessary factor for its biological application [26]”
Additionally, we mainly focused on the concentration of aptamer since the aptamers have the anti-cancer effect. As shown in figure 1, the amount of aptamers on the AuNS was different due to the aptamer’s structure. Thus, the concentration of AuNS in each nanoconstruct was slightly different. To emphasize the aptamer function, we added the aptamer concentration into the x-axis. However, the concentration of AuNS is also an important factor for the reader, we mentioned that cells were treated with different concentrations of AuNS in the figure2 legend. (the concentration of AuNS: anti-cMet = 4.5 nM, anti-nucleolin = 2.5 nM, anti-HER2 = 1.8 nM, respectively). (line 7~8, page 9)
Revised manuscript
“The concentration of AuNS: anti-c-Met = 4.5 nM, anti-nucleolin = 2.5 nM, anti-HER2 = 1.8 nM.”
[Comment #5]
Regarding the in vivo results it would be more significant to show the inhibition of the tumor growth in the main manuscript and not in text. The reverse for the data about vWF and Ki67, if the authors do not want to maintain in the main text, because otherwise there are too many data there.
[Response #5]
Thank you for your feedback. We have relocated the data for inhibition of the tumor growth (Fig.S5) to the main manuscript in Figure 6. We also have referred this in the Results and Discussion section (page 14, line 28 to page 17 line 15 in the revised manuscript)
Revised manuscript
“To further analyze the effect of the AuNS-CN, groups injected with single apt-functionalized AuNSs (AuNS-C, AuNS-N) or a mixture of AuNS-C and AuNS-N (AuNS-C+AuNS-N) were added to our analysis (Fig. 6). Since the intratumoral injection method is advantageous for the tumor targeting efficacy of nanoparticles [29], we applied this method to the mouse model for an accurate comparison of the therapeutic effect of bi-functional AuNS with single aptamer treatment.
From the first day of treatment, the tumor volume measured at one-week interval only tended to decrease in the group treated with AuNS-CN over time (Fig. 6B). In addition, compared to the control group, the tumor volume measured after explant was significantly reduced in the AuNS-C (P = 2.74 × 10-3) and AuNS-CN (P = 4.17 × 10-4) groups, whereas the AuNS-N and AuNS-C + AuNS-N groups did not show a significant difference (Fig. 6C).
H&E and TUNEL staining (Fig. 6D and E) showed that, in the groups treated with AuNS-apts, cellularity was significantly reduced, while the apoptosis of cells was effectively increased compared to the control group (P < 0.0001). In particular, the AuNS-CN group showed an extremely higher apoptotic index value than the AuNS-C, AuNS-N, and AuNS-C+AuNS-N groups (64.36 ± 11.03% in the AuNS-CN group, 31.18 ± 3.86% in the AuNS-C group, 10.83 ± 0.82% in the AuNS-N group, and 11.47 ± 3.56% in the AuNS-C+AuNS-N group).
c-Met and nucleolin apt are known for their ability to inhibit cancer cell proliferation and their anti-angiogenic effect [13, 30]. For these reasons, we performed immunofluorescent staining for vWF (green) and Ki 67(red) to investigate the effect of the AuNS-CN on tumor angiogenesis and proliferation (Fig. 6F). The AuNS-CN group showed the lowest vWF+ cells ratio of 0.56% (±0.65), followed by the AuNS-C group 2.49% (±2.22), AuNS-C+AuNS-N 4.11% (±5.11), AuNS-N 6.69% (±2.62), and the control 21.02% (±17.19) (Fig. 6G). The Ki 67+ cells ratio also showed the lowest value of the AuNS-CN group 0.65% (±0.19) Compared to the AuNS-CN group, single apt-functionalized AuNSs showed approximately 3 times higher, and a mixture of AuNS-C and AuNS-N showed approximately 7 times higher expression (Fig. 6H).”
Figure 6. AuNS-apts inhibited the growth of MKN-45 xenograft tumors in vivo and enhanced the in vivo therapeutic efficacy of combinatorial gold nanoconstructs targeting receptor crosstalk, AuNS-CN. (A) Excised xenograft tumors treated with PBS (control), AuNS-C, AuNS-N, AuNS-C+AuNS-N, and AuNS-CN three times at one-week intervals in Balb/c nude mice (n = 4) injected with 2× 106 MKN-45 cells. (B) The ratio of tumor volume at the indicated time points to initial tumor volume, in each group. (C) Rate of excised tumor volume (4 w) to initial tumor volume in each group. (D) Representative images of H&E and TUNEL staining at excised xenograft tumors treated with PBS (control) and AuNS-apts (AuNS-C, AuNS-N, AuNS-C+AuNS-N, AuNS-CN). In the TUNEL staining images, the green area indicates the distribution of apoptotic cells. Scale bar = 300 µm. (E) Quantification of the apoptotic index (%) of each group. The apoptotic index is the ratio of TUNEL-positive nuclei to total nuclei. (F) Representative images of immunofluorescence-stained vWF and Ki 67. Scale bar = 200 µm. Quantification of: (G) vWF-positive expression areas (%); (H) Ki 67-positive expression areas (%) in the area of total cells to confirm angiogenesis and tumor cell proliferation in xenograft tumors, respectively. *P < 0.01, **P < 0.001, or ***P < 0.0001 compared with the control group. †P < 0.01, ††P < 0.001, or †††P < 0.0001 compared with the AuNS-CN group.
[Comment #6]
Legends should be more detailed.
In the Supplementary (and possibly also in another point) I do not agree with the term “dramatically” in Fig S1. For ISO standard a cell viability around 70% in respect to controls is still accepted as cytocompatibility of the pharmaceutic component, and not as a clear cytotoxic effect (ISO 10993-5:2009—Biological Evaluation of Medical Devices—Part 5: Tests for In Vitro Cytotoxicity. Available online: https://www.iso.org/standard/36406.html.
[Response #6]
We totally agree with Reviewer #1’s point. Therefore, we have excluded the term “dramatically” in the revised Supplementary Manuscript on page 2 as follow:
Revised Supplementary Materials
“Figure. S1. The result of cell viability assay. MKN-45 cells were treated with 250 nM single aptamer-AuNS and bi-functional AuNS with different combinations of aptamers. The viability of MKN-45 cells decreased in AuNS-CN condition as compared with AuNS-NH, AuNS-CH, and single-aptamer-AuNS”
[Comment #7]
Moreover, the English is not that good and it should improved.
[Response #7]
We really appreciated Reviewer #1 for the critical comments. As Reviewer #1 suggested, we have intensively proofread the revised manuscript via the professional agency, Editage. All the errors have been corrected in the revised manuscript.
[Comment #8]
Also the references are not so up to date.
the same ref repeated twice:
- Eder, J. P., G. F. Vande Woude, S. A. Boerner, and P. M. LoRusso. "Novel Therapeutic Inhibitors of the CMet Signaling Pathway in Cancer." Clin Cancer Res 15, no. 7 (2009): 2207-14.
13.Eder, Joseph Paul, George F. Vande Woude, Scott A. Boerner, and Patricia M. LoRusso. "Novel Therapeutic Inhibitors of the C-Met Signaling Pathway in Cancer." Clinical Cancer Research 15, no. 7 (2009): 2207-14.
[Response #8]
Thank you for your comments. We removed duplicate references, 28, and modified references number. We refer to this on page 14, line 42~43 and page 20, line 9 to line 14 of the revised manuscript.
Revised manuscript
“c-Met and nucleolin apt are known for their ability to inhibit cancer cell proliferation and their anti-angiogenic effect [13, 30].”
“Eder, Joseph Paul, George F. Vande Woude, Scott A. Boerner, and Patricia M. LoRusso. "Novel Therapeutic Inhibitors of the C-Met Signaling Pathway in Cancer." Clinical Cancer Research 15, no. 7 (2009): 2207-14.
- Luo, Z., Z. Yan, K. Jin, Q. Pang, T. Jiang, H. Lu, X. Liu, Z. Pang, L. Yu, and X. Jiang. "Precise Glioblastoma Targeting by As1411 Aptamer-Functionalized Poly (L-Gamma-Glutamylglutamine)-Paclitaxel Nanoconjugates." J Colloid Interface Sci 490 (2017): 783-96.
- Tate, Amanda, Shuji Isotani, Michael J. Bradley, Robert A. Sikes, Rodney Davis, Leland W. K. Chung, and Magnus Edlund. "Met-Independent Hepatocyte Growth Factor-Mediated Regulation of Cell Adhesion in Human Prostate Cancer Cells." BMC Cancer 6, no. 1 (2006): 197”
[Comment #9]
page 5, line 4 MKN-46 à MKN-45
MKN45 or MKN-45?
[Response #9]
We apologize for the incorrect labeling again. We have changed all MKN45 to MKN-45 in the revised manuscript as highlighted in yellow. Additionally, we have corrected MKN-46 to MKN-45 in the revised manuscript on page 5, line 4 as follows:
Revised manuscript
“MKN-45 xenografts were generated by subcutaneously injecting 2 × 106 MKN-45 cells into the dorsal flanks of female Balb/C nude mice.”
[Comment #10]
Legend Fig. S5: it must corrected to Balb/c mice
[Response #10]
We apologize for the incorrect information. We have relocated the in vivo data in Fig. S5 to the revised manuscript, and corrected to Balb/c mice in Figure 6 legend of the revised manuscript on page 17 as follows:
Revised manuscript
“(A) Excised xenograft tumors treated with PBS (control), AuNS-C, AuNS-N, AuNS-C+AuNS-N, and AuNS-CN three times at one-week intervals in Balb/c nude mice (n = 4) injected with 2× 106 MKN-45 cells.”
[Comment #11]
These are just examples of mistakes, since I went through to what I remember after reading the paper, but is suggestive of the little accuracy and thus little respect for the reviewers.
[Response #11]
We apologize for the unexpected mistakes in the manuscript. Also, we really appreciate the Reviewer #1 for the precise reviewing the manuscript. We have tried to correct all the mistakes to avoid confusion for the future audience through the revised manuscript (highlighted in yellow). Additionally, we have intensively proofread the manuscript via the professional agency, Editage, and all the suggested corrections have applied in the revised manuscript.

Reviewer 2 Report
In this work, the authors synthesize a nucleolin targeting aptamer (AS1411) and c-Met aptamer (c-Met apt) functionlized star-shaped gold nanostructured (AuNS) therapeutics to achieve combinatorial treatment of cancer. The nanotherapeutics shows synergistic effect which is more effective than that of single aptamer-AuNS at the cellular level. Overall, the manuscript is well presented and provides sufficient data to support their conclusions. However, due to the inorganic nature of AuNS, the biosafety of this nanotherapeutics is a great concern for future clinical application. I strongly suggest that some in vivo experiments should be carried out to investigate the efficacy and biosafety.
- whether the aptamers are stable in body and still targeting to tumor?
- What is the organ distribution of the nanotherapeutics and how to be excreted?
Based on the above questions, I would like to give the decision of major revision.
Author Response
Point-By-Point Response Letter (pharmaceutics-853310)
We are pleased to return the draft of the revised manuscript “Combinatorial Inhibition of Cell Surface Receptors using Dual Aptamer-functionalized Nanoconstructs for Cancer Treatment” by Lee et al., for consideration of publication in Pharmaceutics, based on the responses to the Reviewer 1 and 2. They found our work of interest and significance in this field, and we received an essentially positive response regarding our manuscript. We are pleased to add a number of substantial revisions requested by the Reviewers that address the concerns and comments in detail, and we believe that we have now supported our model with extensive and detailed data.
The main point of the paper is the dual-targeting strategy to enhance anti-cancer efficacy via synergistic proximity interaction of therapeutics with two receptor proteins. We have provided most of the extensive prepared additional revision requested to more vigorously address the list of mechanistic questions suggested by Reviewers (please see revised Fig. 6, Fig S1 and Fig S3). We believe that this supports the therapeutic value of targeting the cross-talk between c-Met and nucleolin by the bispecific nanocarrier densely grafted with anti-c-Met and –nucleolin aptamer increasing the local concentration of aptamers at the target sites. In the following, we addressed all the comments made by the reviewers on our manuscript (pharmaceutics-853310) point by point. In response to each raised point, we explained the corresponding changes we have made in the revised manuscript (highlighted in yellow).
Reviewer #2
Comments and Suggestions for Authors:
In this work, the authors synthesize a nucleolin targeting aptamer (AS1411) and c-Met aptamer (c-Met apt) functionlized star-shaped gold nanostructured (AuNS) therapeutics to achieve combinatorial treatment of cancer. The nanotherapeutics shows synergistic effect which is more effective than that of single aptamer-AuNS at the cellular level. Overall, the manuscript is well presented and provides sufficient data to support their conclusions. However, due to the inorganic nature of AuNS, the biosafety of this nanotherapeutics is a great concern for future clinical application. I strongly suggest that some in vivo experiments should be carried out to investigate the efficacy and biosafety.
[Comment #1]
1.whether the aptamers are stable in body and still targeting to tumor?
[Response #1]
We really appreciated Reviewer #2 for the critical question. The aptamers consisted of nucleic acid should have a strong negative charge that can prevent particle aggregation in physiological conditions. According to the reference, the aptamer-functionalized AuNS was stable in the blood that was proved using by UV-Vis spectrometer. (Nanomedicine. 2015, 11(3): 671–679). As the reviewer commented, the stability of nanoconstructs in the body is an important point for biological application. Thus we mentioned the stability of AuNS in vivo and cited the reference. (line 42~43, page 6).
Figure S2 in the reference (Nanomedicine. Teri W. Odom et al. 2015, 11(3): 671–679): Colloidal stability of the nanoconstructs in PBS and whole blood. LSP resonance of Apt-AuNS in PBS was ca. 800 nm, which indicated stability of Apt-AuNS in PBS. Similarly, the LSP of nanoconstructs in whole blood (red line) also showed a resonance at 800 nm with slight broadening due to protein adsorption; no aggregation was observed in whole blood.
Revised manuscript
“As shown in Fig. 1C, all nanoconstructs had a negatively charged surface, which contributed to their stable suspension in the cell culture medium and in whole blood via repulsion between nanoparticles [26].”
Importantly, the aptamers on the surface of AuNS are more stable in physiological condition compared with the free aptamer molecules. The negatively charged aptamer-AuNS recruits salt ion resulting in formation high local salt concentration around the particle surface (Nano Lett. Chad A. Mirkin et al. 2009, 9(1): 308–311). As a result, the nuclease enzymatic activity decreases at the AuNS surface, which contributes improvement of aptamer stability in the body.
This point is a strong advantage for aptamer functionalization on the AuNS, thus we add this information into the introduction part (line 38~44, page17, reference 32).
Scheme 1 in the reference (Nano Lett. 2009, 9(1): 308–311), Proposed Mechanism for Polyvalent Nanoparticle-Induced DNA Stability. Enzyme catalyzed DNA hydrolysis is modeled in two steps: enzyme association, and DNA hydrolysis. The high surface-charge resulting in increased associated salts is postulated to be the origin of DNA stability and the slower enzymatic hydrolysis.
Revised manuscript
“Importantly, the aptamers on the surface of AuNS are more stable under physiological conditions than free aptamer molecules. The negatively charged aptamer-AuNS recruits salt ions, resulting in a high local salt concentration around the particle surface [32]. As a result, the nuclease enzymatic activity decreases at the AuNS surface, which contributes to an improved aptamer stability in the body. The immobilization of aptamers onto the surface of AuNS at a high density was found to improve the anti-cancer effect, as well as increase the targeting efficacy, simultaneously, for the application of therapeutic aptamers in vivo.”
[Comment #2]
2.What is the organ distribution of the nanotherapeutics and how to be excreted?
[Response #2]
We really appreciated Reviewer #2 for the constructive question. In our study, we treated the nanoparticle to mouse through by intratumoral injection method. The amount of gold significantly increased after 24 h injection at the tumor site for an intratumorally injected mouse. In particular, the dose of gold at tumor was 5-times higher than the dose at kidney. However, the tail vein injection approach showed the highest dose of gold at liver, kidney, and spleen. (PNAS, Oleg A. Andreev et al. 2013 110 (2) 465-470). This indicates that the method of intratumoral injection is advantageous for tumor targeting efficacy of nanoparticle. Thus, we expect that the amount of nanotherapeutics is also the highest at the tumor in our study that is almost the same to the trend of gold distribution in reference. We mentioned why we applied the intratumoral injection method in the revised manuscript. (line 28~30, page 14, reference 29)
Fig. 3 in the reference (PNAS, 2013 110 (2) 465-470) ICP-MS analysis of the amount of gold in the excised tissues. Either target the entire mass. Direct injection resulted in accumulation a single intratumoral injection of gold-pHLIP or gold nanoparticles (A; 20 μM, 50 μL) or two i.v. injections of gold-pHLIP, gold-K-pHLIP, and gold (B; 20 μM, 150 μL each) were given to mice bearing s.c. tumors. Organs were collected at 24 h after the last injection. Organ uptakes of gold via intratumoral and IV injections are shown in A and B, respectively
Revised manuscript
“Since the intratumoral injection method is advantageous for the tumor targeting efficacy of nanoparticles [29], we applied this method to the mouse model for an accurate comparison of the therapeutic effect of bi-functional AuNS with single aptamer treatment.”

Round 2
Reviewer 1 Report
Now that the manuscript has been revised, I can read it better, but I still see some points that must be improved.
Fig. 1D. I still do not agree about naming the aptamer-NP as they are written (see my comments on my previous revision). Anyway, at least eliminate the terms anti in the different acronyms on the left boxes of the table.
Fig. 3A. From the legend I understand that the simple aptamers (and not the functionalized nanoparticles) were tested. In the top panel (Nucleolin + c-MET): there is significant reduction in cell viability of MKN-45 cells. This should be taken into account. Legend of the panel at the bottom: MKN54 should be MKN-45.
Fig. 3B. Since the amounts of the simple aptamers are reported in Fig. 3A, it would help the reader to know also from the legend the amounts of the aptamers and of the aptamers loaded on the nanoparticles used in the experiments and thus this information should be added in the Legend of the figure.
The experiments reported in Fig 3C and 3D are aimed to understand the interaction of the molecules (i.e. the receptors) recognized by aptamers, besides confirming that the latter react with the receptors, as already shown in the experiments of immunofluorescence (Fig. 2A). So first of all the terms correlation and correlated in the legend of Fig 3 are not appropriate and do not means anything in this context. The same in the Conclusions. The right word is interaction, and this was shown at molecular level by means of pull-down experiments (Fig. 3D) or co-immunoprecipitation experiments (Fig. 3E). Taking into account my comments the authors should write better all this part.
As it appears from the Figs. 3D and 3E the Nucleolin appear to interact with Met receptor as shown both in pull-down and co-immunoprecipitation experiments. The reverse is not observed, i.e. pulled-down Met receptor does not interact with Nucleolin. The experiment would be complete if anti-Met antibodies were used to co-immunoprecipitate Nucleolin (i.e., the same type of experiment as in Fig 3E for Nucleolin). If confirmed, this different results in respect to the two receptor should be discussed.
page 14, line 42: the sentence is non corrected. c-Met and nucleolin have a strong pro-tumor effect. Their inhibition has anti-tumor and anti-angiogenic effect.
The data about Ki67 is strange. How can the authors comment about the fact that the administration of the two singly functionalized aptamers nanoparticles together are less effective than the functionalized by the c-Met alone, which in general gave better results, as therapeutic, in these studies?
Ref 31. at the beginning Tate Amanda, (no coma in between)
Fig. S3 In the titles of the graphs: both and only are pleonastic and should be eliminated.
Once the authors have answered to the different points, the paper could be published. (one experiment is still asked for)
Author Response
Point-By-Point Response Letter (pharmaceutics-853310)
We are pleased to return the draft of the revised manuscript “Combinatorial Inhibition of Cell Surface Receptors using Dual Aptamer-functionalized Nanoconstructs for Cancer Treatment” by Lee et al., for consideration of publication in Pharmaceutics. According to the Reviewer’s new suggestions and comments, we made our efforts to change descriptions to improve and strengthen the points in the revised manuscript. We are very pleased to receive positive and constructive Reviewer’s comments, and provide a detailed draft of our responses to the questions raised by the Reviewers.
Most of all, we have done a complete overhaul of the manuscript including significant restructuring to point out common mechanistic themes as well as putting current knowledge more into context. We have responded in full to all the Reviewer’s comments (please see below for point-by-point response). Hope the Reviewers like the revised manuscript.
We look forward to hearing good news from you. We are confident that this manuscript will have high impact in several fields. Thank you for your consideration.
Reviewer #1
Now that the manuscript has been revised, I can read it better, but I still see some points that must be improved.
[Comment #1]
Fig. 1D. I still do not agree about naming the aptamer-NP as they are written (see my comments on my previous revision). Anyway, at least eliminate the terms anti in the different acronyms on the left boxes of the table.
[Response #1]
We appreciate Reviewer#1 for the concerns about naming the construct. As Reviewer #1 suggested, we have eliminated the term “anti” in Fig 1D on page 7. Additionally, we changed the figure 2 on page 9.
Revised manuscript
[Comment #2]
Fig. 3A. From the legend I understand that the simple aptamers (and not the functionalized nanoparticles) were tested. In the top panel (Nucleolin + c-MET): there is significant reduction in cell viability of MKN-45 cells. This should be taken into account. Legend of the panel at the bottom: MKN54 should be MKN-45.
[Response #2]
We apologize for the confusion. The reagent used in Fig. 3A was the functionalized nanoparticles, not the simple aptamer. We have corrected labeling AuNS-CN, AuNS-HN, AuNS-CH in Figure 3A and included the additional description in the revised manuscript on page 11, line 4~ page12, line 5 to clarify the experimental set-up and the result as follow:
Revised manuscript
Figure 3. Combinatorial treatment of cells with the bi-functional-AuNS. (A) Cell viability using MTS. The three cell lines were treated with nanoconstructs that were functionalized by a different combination of aptamers. CN: AuNS functionalization with anti-c-MET and -nucleolin aptamer, HN: AuNS functionalization with anti-HER2 and –nucleolin aptamer, and CH: AuNS functionalization with anti-c-MET and –HER2 aptamer. (B) The result for viability after cells treatment with 250 nM free aptamer or 250 nM aptamer-functionalized AuNS (AuNS Conc.1.125 nM for c-MET, 0.625 nM for nucleolin, 1 nM for CN, respectively).
[Comment #3]
Fig. 3B. Since the amounts of the simple aptamers are reported in Fig. 3A, it would help the reader to know also from the legend the amounts of the aptamers and of the aptamers loaded on the nanoparticles used in the experiments and thus this information should be added in the Legend of the figure
[Response #3]
We appreciate your critical comment. As Reviewer #1 suggested, we added the information of the concentration for aptamer and nanoparticle in the legend of the figure on page 11, line 8~10.
Revised manuscript
Figure 3. Combinatorial treatment of cells with the bi-functional-AuNS. (A) Cell viability using MTS. The three cell lines were treated with nanoconstructs that were functionalized by a different combination of aptamers. CN: AuNS functionalization with anti-c-MET and -nucleolin aptamer, HN: AuNS functionalization with anti-HER2 and –nucleolin aptamer, and CH: AuNS functionalization with anti-c-MET and –HER2 aptamer. (B) The result for viability after cells treatment with 250 nM free aptamer or 250 nM aptamer-functionalized AuNS (AuNS Conc.1.125 nM for c-MET, 0.625 nM for nucleolin, 1 nM for CN, respectively).
[Comment #4]
The experiments reported in Fig 3C and 3D are aimed to understand the interaction of the molecules (i.e. the receptors) recognized by aptamers, besides confirming that the latter react with the receptors, as already shown in the experiments of immunofluorescence (Fig. 2A). So first of all the terms correlation and correlated in the legend of Fig 3 are not appropriate and do not means anything in this context. The same in the Conclusions. The right word is interaction, and this was shown at molecular level by means of pull-down experiments (Fig. 3D) or co-immunoprecipitation experiments (Fig. 3E). Taking into account my comments the authors should write better all this part.
As it appears from the Figs. 3D and 3E the Nucleolin appear to interact with Met receptor as shown both in pull-down and co-immunoprecipitation experiments. The reverse is not observed, i.e. pulled-down Met receptor does not interact with Nucleolin. The experiment would be complete if anti-Met antibodies were used to co-immunoprecipitate Nucleolin (i.e., the same type of experiment as in Fig 3E for Nucleolin). If confirmed, this different results in respect to the two receptor should be discussed.
[Response #4]
We appreciate Reviewer#1 for the critical comments. As Reviewer#1 suggested, we changed the inappropriate expression in the abstract and main text and added the new description in the revised manuscript on page 1 line 32-35, page 12 line 2-5, and page 15 line 23 as follows:
Revised manuscript
page 1 line 32-35
Our findings pave the way for the development of an effective combinatorial treatment based on nanoconstruct-mediated interaction between receptors.
Keywords: Combinatorial treatment, Aptamer, Gold Nanoconstructs, Surface Receptor, Receptor Interaction
page 12 line 2-5
(E) Immune precipitation upon using the anti-nucleolin antibody to determine the interaction of nucleolin with c-Met. Ab-MMP is an antibody-functionalized micromagnetic particle (MMP) and MMP is a bare particle (without antibody). The immunoblotting data shows that nucleolin interacts with c-Met.
page 15 line 23
Despite the significance of targets in cancer therapy, there has been no previous evidence demonstrating an interaction between c-Met and nucleolin via nanoconstruct.
[Comment #5]
page 14, line 42: the sentence is non corrected. c-Met and nucleolin have a strong pro-tumor effect. Their inhibition has anti-tumor and anti-angiogenic effect.
[Response #5]
We apologize for the confusion caused by the incorrect expression on page 14, line 42. We have changed the term as follows:
Revised manuscript
“Anti-c-Met and anti-nucleolin apt are known for their ability to inhibit cancer cell proliferation and their anti-angiogenic effect.”
[Comment #6]
The data about Ki67 is strange. How can the authors comment about the fact that the administration of the two singly functionalized aptamers nanoparticles together are less effective than the functionalized by the c-Met alone, which in general gave better results, as therapeutic, in these studies?
[Response #6]
Thank you for your critical comments. In order to equalize the dose of the loaded total functionalized aptamers, the amount of each functionalized aptamer in the AuNS-C+AuNS-N group was loaded in a smaller amount than the groups loaded with single aptamer. Ki67 data is a fragmentary indicator of the proliferation of tumor cells. Because the effects of anti-c-Met and ani-nuculolin aptamers on the proliferation of the tumor cells, and the effective dose could be different, the use of two singly functionalized aptamers nanoparticles together may be less effective than using a single aptamer.
[Comment #7]
Ref 31. at the beginning Tate Amanda, (no coma in between)
[Response #7]
We apologize for the typo. We have eliminated the coma in Ref31 on page 20, line 12 as follows:
Revised manuscript
- Tate Amanda, Shuji Isotani, Michael J. Bradley, Robert A. Sikes, Rodney Davis, Leland W. K. Chung, and Magnus Edlund. "Met-Independent Hepatocyte Growth Factor-Mediated Regulation of Cell Adhesion in Human Prostate Cancer Cells." BMC Cancer 6, no. 1 (2006): 197.
[Comment #8]
Fig. S3 In the titles of the graphs: both and only are pleonastic and should be eliminated.
[Response #8]
We appreciate Reviewer#1 for the critical comment. As Reviewer#1 suggested, we have eliminated the term “both” and “only” in Fig. S3 in the revised Supplementary Materials on page 3 as follows:
Revised Supplementary Materials

Reviewer 2 Report
The authors have addressed the concerns raised by the reviewers.
Author Response
Point-By-Point Response Letter (pharmaceutics-853310)
We are pleased to return the draft of the revised manuscript “Combinatorial Inhibition of Cell Surface Receptors using Dual Aptamer-functionalized Nanoconstructs for Cancer Treatment” by Lee et al., for consideration of publication in Pharmaceutics. According to the Reviewer’s new suggestions and comments, we made our efforts to change descriptions to improve and strengthen the points in the revised manuscript. We are very pleased to receive positive and constructive Reviewer’s comments, and provide a detailed draft of our responses to the questions raised by the Reviewers.
Most of all, we have done a complete overhaul of the manuscript including significant restructuring to point out common mechanistic themes as well as putting current knowledge more into context. We have responded in full to all the Reviewer’s comments (please see below for point-by-point response). Hope the Reviewers like the revised manuscript.
We look forward to hearing good news from you. We are confident that this manuscript will have high impact in several fields. Thank you for your consideration.
Reviewer #2
Comments and Suggestions for Authors:
The authors have addressed the concerns raised by the reviewers.
[Response]
We appreciate Reviewer #2 for the constructive discussion regarding the biosafety and compatibility of the nanoconstruct. We believe that all our results are novel and will be of great general interest to the broad scientific community working in all fields of biological and medical sciences.
